# Active diffusion and advection in *Drosophila* oocytes result from the interplay of actin and microtubules

Maik Drechsler[1], Fabio Giavazzi [2], Roberto Cerbino [2] & Isabel M. Palacios[1,3]

Transport in cells occurs via a delicate interplay of passive and active processes, including diffusion, directed transport and advection. Despite progress in super-resolution microscopy, discriminating and quantifying these processes is a challenge, requiring tracking of rapidly moving, sub-diffraction objects in a crowded, noisy environment. Here we use differential dynamic microscopy with different contrast mechanisms to provide a thorough characterization of the dynamics in the *Drosophila* oocyte. We study the movement of vesicles and the elusive motion of a cytoplasmic F-actin mesh, a known regulator of cytoplasmic flows. We find that cytoplasmic motility constitutes a combination of directed motion and random diffusion. While advection is mainly attributed to microtubules, we find that active diffusion is driven by the actin cytoskeleton, although it is also enhanced by the flow. We also find that an important dynamic link exists between vesicles and cytoplasmic F-actin motion, as recently suggested in mouse oocytes.

[1] Department of Zoology, University of Cambridge, CB2 3EJ Cambridge, UK. [2] Department BIOMETRA, University of Milan, 20090 Segrate, Milan, Italy. [3] School of Biological and Chemical Sciences, Queen Mary University of London, Mile End Road, London E1 4NS, UK. Maik Drechsler and Fabio Giavazzi contributed equally to this work. Roberto Cerbino and Isabel M. Palacios jointly supervised this work. Correspondence and requests for materials should be addressed to R.C. (email: roberto.cerbino@unimi.it) or to I.M.P. (email: mip22@cam.ac.uk)

The distribution and organization of cytoplasmic content, like proteins, nucleic acids, and organelles require the combined action of passive and active biophysical processes. Thermal-based diffusion is not sufficiently fast and effective in redistributing large organelles, like vesicles, within the crowded and viscous environment found inside a cell[1]. Active transport mechanisms mitigate the ineffectiveness of thermal diffusion. Motor proteins carry attached cargos (e.g., vesicles) along cytoskeletal filaments, which act as tracks for directed transport across the cell[2]. In larger cells, it is likely that the transport of such cargoes also causes a large-scale net flow, known as cytoplasmic streaming[3,4]. As a result of the viscous drag, caused by a translocating motor, streaming leads to a circulation of the cytoplasm and its efficient remixing[5,6]. Finally, ATP-dependent processes are responsible for the presence of random force fluctuations within the cytoplasm, whose effects lead to the displacement of tracer particles in a diffusive-like manner—named active diffusion—that is more efficient than thermal-based diffusion[7–11]. Understanding the details of the subtle interplay between all these processes is a demanding task, due to the many time- and length-scales and the multiple molecular pathways involved.

Here we investigate the interactions between different motion mechanisms in *Drosophila* oocytes. *Drosophila* oogenesis is well studied genetically and oocytes can be probed with a variety of chemical treatments and microscopic tools. In the oocyte, microtubules and kinesin-1 are essential for both cargo transport and cytoplasmic streaming[12,13]. At mid-oogenesis (stage 9, st9), the topology and speed of cytoplasmic flows directly correlates with the topology of the microtubule cytoskeleton and the speed of kinesin, respectively[14]. In addition, a cytoplasmic network of actin filaments (F-actin)—known as the actin mesh—acts as a negative regulator of the microtubule/kinesin-dependent flow[15]. However, much remains to be uncovered about the interplay between the actin mesh and the microtubule cytoskeleton in regulating the motion of ooplasmic material. The recent discovery of a link between cytoplasmic actin and vesicle dynamics in mouse oocytes already suggests a close relationship between vesicle transport, motor activity and the actin cytoskeleton[16]. However, it remains unclear whether this observation represents a general feature of cells.

To gain insight on these issues we developed a methodology to simultaneously probe the dynamics of both, cytoplasmic F-actin and vesicles in wild-type oocytes, as well as in oocytes with aberrant cytoskeletons and different cytoplasmic streaming conditions. We did so by combining particle image velocimetry (PIV) with differential dynamic microscopy (DDM)[17,18]. In contrast to PIV, DDM probes the sample dynamics not in the direct space, but in the Fourier domain, with the bonus of being able to analyze densely distributed objects, whose size is well below the diffraction limit of the microscope, and in the presence of substantial amounts of noise. DDM can be employed with different imaging mechanisms, providing information on various dynamic structures. These properties are used here for the first time to characterize the crowded interior of a living cell. This is done by combining confocal imaging of labeled F-actin, with simultaneous differential interference contrast (DIC) imaging of unlabeled vesicles.

Using chemical and genetic manipulations we show that vesicle dynamics result from two contributions: a persistent ballistic motion due to cytoplasmic flows and a diffusion process of active nature. We found that a similar combination of ballistic and diffusive movements also captures the motility of F-actin, showing a strong correlation between the motion of the actin network and the DIC vesicles. This result provides an important link between the active diffusion of vesicles and the underlying fluctuating non-equilibrium actin network, of which we quantify the overall dynamics by focusing on the diffusive-like component.

Finally, we demonstrate that, in our system, active diffusion constitutes an ATP-dependent process, with at least two distinct ingredients. The actin mesh itself seems to be a major source of active diffusion. However, we also find that microtubules enhance active diffusion, and only depletion of both cytoskeletons results in the abrogation of this random motion. We demonstrate that microtubules substantially contribute to the diffusive motion of both vesicles and the cytoplasmic actin network.

In summary, our work sheds new light on the dynamic interplay between ATP-dependent forces and cytoplasmic mechanics to regulate intracellular motility. We show that: (1) the major ATP-dependent entities responsible for advection and active diffusion are the microtubule and cytoplasmic F-actin networks, and (2) an important dynamic link exists between vesicles and cytoplasmic F-actin motion.

From a more methodological perspective, we establish DDM as a powerful tool for all biologists interested in motion and in the rearrangement dynamics of different structures, as it allows to extract a robust quantitative information even in conditions where more traditional image processing methods fail.

## Results

**Vesicle dynamics consist of persistent and diffusive motion.** Asymmetric localization of developmental determinants by motors in st9 oocytes is a key event for the specification of the body axes of the embryo[19]. In addition, the translocation of cargos by kinesin-1 induces cytoplasmic flows. These flows can be measured by PIV, using endogenous vesicles as tracer particles[14]. However, while PIV gives an accurate description of flow velocities and topology, it is unsuitable to describe non-persistent, diffusive motion. Therefore, we used DDM to monitor and characterize cytoplasmic movements in more detail (Fig. 1)[18,20]. Analyzing DIC time-lapse movies of oocytes by DDM (DIC–DDM) unveiled a complex dynamic behavior, which is not fully captured by PIV. In fact, DIC–DDM analysis shows that ooplasmic vesicles move in a ballistic, persistent, as well as in a random, diffusive manner (Fig. 1).

In DDM experiments, the information about the sample dynamics is encoded in the intermediate scattering function (ISF) $f(q, \Delta t)$, which describes the relaxation of density fluctuations with wave vector $q$ as a function of the delay time $\Delta t$ (Fig. 1c)[18,21]. Initial attempts of fitting our experimental ISFs to the prediction of an advection model, inspired by PIV results, failed. Our ISFs were clearly suggesting a more complex dynamic. We obtained a successful description by using a simple advection–diffusion model, given by

$$f(q, \Delta t) = P_0(\Gamma_1(q)\Delta t)e^{-\Gamma_2(q)\Delta t}, \qquad (1)$$

in which, in addition to a directional, ballistic motion with rate $\Gamma_1(q)$, vesicles are also subjected to a random, diffusive motion with rate $\Gamma_2(q)$. We found a good agreement between our model and the experimental data by considering that each vesicle bears the same diffusivity $D_{ves}$ ("ves" for vesicles), but a different velocity drawn from a prescribed probability distribution function, whose Fourier transform $P_0$ appears in Eq. (1) (see "Methods"). In fact, once the average vesicle speed $v_{ves}$ is calculated, one has $\Gamma_1(q) = v_{ves}q$ and $\Gamma_2(q) = D_{ves}q^2$. Fitting the experimental ISFs (Fig. 1c) to Eq. (1), confirmed the validity of our model and allowed us to simultaneously determine $\Gamma_1$ and $\Gamma_2$ for each $q$ (Fig. 1d), both exhibiting the expected ballistic or diffusive scaling, respectively. By repeating this analysis on seven cells (with an average of 1,500 vesicles per cell contributing to the DIC–DDM signal) we obtained $v_{ves} = 36 \pm 15$ nm s$^{-1}$ and

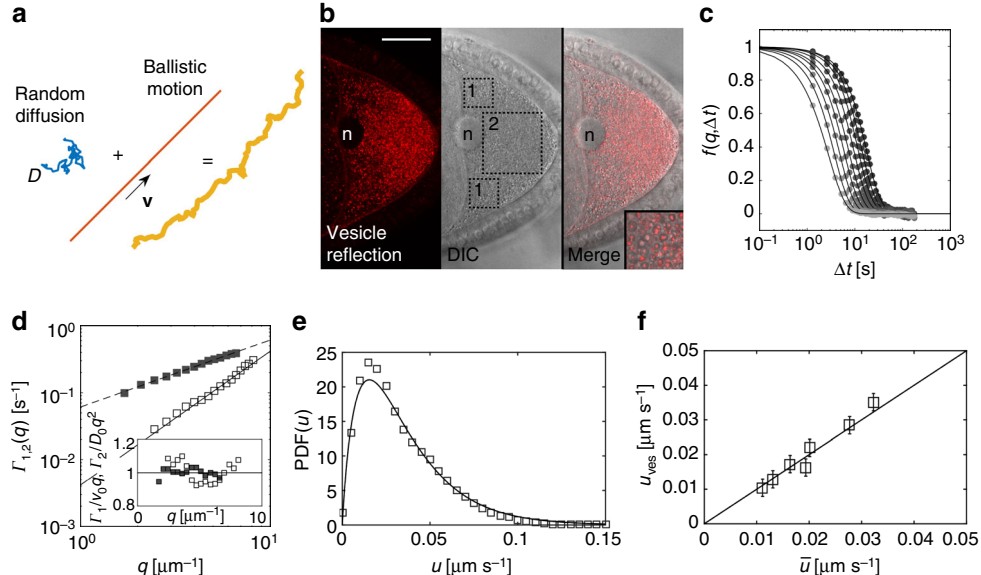

**Fig. 1** Movement of cytoplasmic vesicles consists of persistent and diffusive motions. **a** Diagram, depicting the motion of cytoplasmic particles by random diffusion, described by a characteristic diffusion coefficient $D$ (blue), and a persistent, ballistic motion, described by a characteristic velocity $v$ (red). Both together result in the persistent, erratic movement of tracer particles observed in the cell (orange). **b** Confocal image showing reflection of vesicles in a st9 oocyte (left panel), and the same vesicles imaged by DIC microscopy (middle panel). The merge of both signals reveals that the reflective signal co-localizes with the center of vesicles (right panel, inset). The boxed areas indicate anterior (1) and posterior (2) regions of interest (ROI), used to refer positional differences of diffusivity and streaming velocities. As expected, flow velocities in the anterior part of the cell are markedly faster than posteriorly (anterior $v_{ves} = 45 \pm 10$ nm s$^{-1}$ vs. posterior $v_{ves} = 28 \pm 5$ nm s$^{-1}$). n nucleus. Scale bar is 30 μm. **c** Intermediate scattering functions (ISF) $f(q,\Delta t)$, obtained from DIC–DDM analysis for different wave vectors $q$ in the range 1.8 μm$^{-1} < q < 8$ μm$^{-1}$. Continuous lines are best fits to Eq. (1). **d** Decorrelation rates $\Gamma_1(q)$ (solid black boxes) and $\Gamma_2(q)$ (open boxes) obtained from the fit of the ISF, represented in **c**, plotted against the wave vector $q$. $\Gamma_1(q)$, which accounts for the ballistic contribution to the motion of the vesicles, exhibits a linear scaling $\Gamma_1(q) = v_{ves}q$ (dashed line), while $\Gamma_2(q)$, which describes a diffusive-like relaxation process, is well fitted to a quadratic law $\Gamma_2(q) = D_{ves}q^2$ (continuous line). In the inset, the ratios $\Gamma_1(q)/(v_{ves}q)$ (solid black boxes) and $\Gamma_2(q)/(D_{ves}q^2)$ (open boxes) are reported, showing no systematic deviations from the expected value (1) and an overall moderate fluctuation (<10%). **e** Probability distribution function (PDF) of the 2D streaming speed in a single cell, as obtained from PIV analysis (open boxes) and as reconstructed from DIC–DDM analysis (solid line). **f** Comparison of 2D mean streaming speeds obtained by DIC–DDM ($u_{ves}$) and PIV ($\bar{u}$) for each cell tested. Both values are in good agreement (solid line represents $u_{ves} = \bar{u}$), validating DIC–DDM as reliable tool to quantify cytoplasmic flows. Error bars are standard deviation

$D_{ves} = (3 \pm 1)10^{-3}$ μm$^2$ s$^{-1}$, where the uncertainty is taken as the standard deviation of the population (Table 1).

To analyze motion, three different regions of interest (ROIs) were taken in each cell (Fig. 1b). As the final result, we present an average over all ROIs. However, it should be noted that the motion at the posterior of the cell displays a slower ballistic component and a more pronounced diffusive-like behavior than the motion in anterior ROIs. We find for the posterior ROI $v_{ves}^{(p)} = 28 \pm 5$ nm s$^{-1}$ and $D_{ves}^{(p)} = (3.5 \pm 1) \times 10^{-3}$ μm$^2$ s$^{-1}$, while for the average of two ROIs, symmetrically located in the anterior of the cell, we get $v_{ves}^{(a)} = 45 \pm 10$ nm s$^{-1}$ and $D_{ves}^{(a)} = (2.3 \pm 1) \times 10^{-3}$ μm$^2$ s$^{-1}$. This anterior–posterior velocity gradient is in agreement with previous flow measurements, as well as the anterior–posterior gradient displayed by the microtubule network[12–14,22–25].

The ratio $r = 6D_{ves}/v_{ves} \simeq 500$ nm corresponds to a characteristic length scale that separates two distinct regimes: over distances larger than $r$, advection is the most efficient transport mechanism, while on smaller length scales diffusion prevails. Of note, in our case $r$ is roughly of the order of the vesicle radius. Therefore, it is not surprising that PIV, which operates over a coarse-grained grid with a resolution larger than the size of the tracers, fails to capture the erratic, small scale, diffusive movement. On the other hand, PIV can efficiently measure the persistent, large-scale motion of the vesicles, which can be used to validate our DIC–DDM approach. To this aim, we analyzed the same DIC movies by PIV (Fig. 1e, f). The comparison between DIC–DDM and DIC–PIV shows that both methods reveal the same quantitative description of flows. With DIC–PIV we find a mean vesicle speed of $\bar{u} = 20 \pm 6$ nm s$^{-1}$ that is in good agreement with the 2D projection $u_{ves}$ of the 3D speed $v_{ves}$ obtained by DIC–DDM, where $u_{ves} = 0.56$ and $v_{ves} = 20 \pm 7$ nm s$^{-1}$ (Fig. 1e, f). Comparable information can also be extracted by PIV analysis of movies of images from the light reflected by vesicles at 561 nm[14]. We find that each DIC vesicle reflects light at 561 nm from its center (Fig. 1b), and therefore, it is not surprising that they display the same motion (Supplementary Fig. 1). More importantly, the nearly identical velocity values obtained by DIC–PIV and DIC–DDM support the hypothesis that the persistent component of the vesicle's motion captured by DIC–DDM corresponds to the microtubule-dependent flow, previously described by PIV only[14]. However, compared to reflection imaging, DIC has three advantages for general studies on motion of cytoplasmic components: the focal plane is thicker; there is no need to use a specific laser line, which allows the use of a larger number of fluorophores; it can be applied to any cell type.

In conclusion, we have shown that DDM can quantitatively separate the persistent, ballistic motion from the random, diffusive-like movement experienced by the same set of vesicles. Importantly, this is obtained through a simple and fully automated procedure, that does not require the accurate localization, tracking and trajectory reconstruction of single tracer particles on which direct-space methods typically rely[26–28]. This makes the DDM results particularly robust and user-independent, as they are not affected by the selection bias or by

**Table 1 Summary of motion data**

| Description | $D_{ves}$ (x$10^{-3}$ µm² s⁻¹) | $D_{act}$ (x$10^{-3}$ µm² s⁻¹) | $v_{ves}$ (nm s⁻¹) | $v_{act}$ (nm s⁻¹) | Genotype |
|---|---|---|---|---|---|
| control cells[1] | 3±1 | 6±1.5 | 36±15 | 36±15 | *sqh*-UTRN.GFP |
| no microtubules[1] | 1.4±0.5 | 2.8±1 | nd | nd | *sqh*-UTRN.GFP + colchicine |
| ATP depleted[1] | 0.1±0.5 | 0.14±0.8 | nd | nd | *sqh*-UTRN.GFP + NaN₃, 2-*D* deoxyglucose |
| no cytoplasmic F-actin[1] | nd | -- | 150±70 | -- | *spire*[1]/Df(2L)Exel6046; *sqh*-UTRN.GFP |
| no cytoplasmic F-actin, no microtubules[1] | 0. 6±0. 2 | -- | nd | — | *spire*[1]/Df(2L)Exel6046; *sqh*-UTRN.GFP + colchicine |
| control cells[2] | 2±0.5 | 3.5±1 | 13±3 | 9±5 | *nos*-Gal4,*sqh*-UTRN.GFP>RFP |
| more cytoplasmic F-actin[2] | 1.4±0.5 | 2.3±0.8 | nd | nd | *nos*-Gal4,*sqh*-UTRN.GFP>Spire.B.tdTomato |
| | 1.2±0.7 | 2±1 | nd | nd | *nos*-Gal4,*sqh*-UTRN.GFP>Capu.tdTomato |
| reduced Shot[2] | 4.6±2 | -- | 11±7 (all) 15.8±3 (no 0)[3] | -- | *nos*-Gal4,*sqh*-UTRN.GFP>*shot*.RNAi |
| F-actin binding of Shot disrupted[1] | 4±1 | -- | 26±26 (all) 43±17 (no 0)[3] | -- | *shot*[kakP1] germ line clones |
| Arp2/3 complex aberrant[1] | 1.5±0.5 | -- | 12±8 | — | *Arpc1*[Q25sd] germ line clones |
| **description** | **$D_{mito}$ (x$10^{-3}$ µm² s⁻¹)** | | **$v_{mito}$ (nm s⁻¹)** | | **genotype** |
| mitochondria | 1.8±0.5 | -- | 15±4 | — | *sqh*-mito.YFP |

Values for diffusive (diffusion coefficient *D*) and ballistic (velocity *v*) motions obtained by DIC–DDM (vesicles, ves) or Con-DDM (actin, act; mitochondria, mito) in different genotypes and under different conditions of treatment. Note that expression of RFP by *nos*-Gal4 already affects the motion in oocytes. Therefore, all over-expression and RNAi experiments need to be compared to a separate control (gray boxes)
nd not detectable;—structure not present or labeled; all values given are mean values ± standard deviation 1—compare to control 1 (white shading); 2—compare to control 2 (gray shading); 3—the given genotype resulted in some cells without any detectable flow, average values are given with those cells included (all) or excluded (no 0)

the strong dependence on external parameters that are often associated with manual and automated particle tracking. In addition, our method is non-invasive, as no tracer particles need to be injected into the cells, and we do not need to apply external forces to characterize the motion of cytoplasmic components.

Our approach can be of use in a variety of biological problems, involving the characterization of the motion and the restructuring dynamics of different cytoplasmic components, as it provides a detailed and statistically robust description, even in conditions where single particles and trajectories cannot even be resolved or identified.

**The motility of cytoplasmic F-actin and vesicles correlates**. The cytoplasm constitutes a densely packed environment, containing not only organelles, but also highly dynamic F-actin. After successfully using DIC–DDM to quantify vesicle motion, we applied it to monitor the motility of the F-actin network traversing the ooplasm. This task is made difficult by the small size of the filaments, their fast and random movement, their crowding, and finally a low signal-to-noise ratio[8,29–31].

To capture the overall cytoplasmic dynamics, we imaged vesicles and F-actin simultaneously (Fig. 2 and Supplementary Fig. 2). In st9 oocytes, a three-dimensional F-actin network traverses the ooplasm[15]. This actin mesh is formed by the cooperative activity of two nucleators, Spire and the formin Capuccino[15,32,33]. Staining fixed oocytes using phalloidin shows the presence of intertwined actin filaments, as well as F-actin rich foci (Fig. 2b, b′ and Supplementary Fig. 2c). The dynamic behavior of this structure is yet unknown. Fluorescently labeling actin in living cells is challenging and yeast formins reject tagged actin monomers[34]—a fact we could confirm in *Drosophila* oocytes. Fluorophore-tagged Act5C (one out of six *Drosophila* actin proteins, and the only one ubiquitously expressed) becomes

incorporated into cortical F-actin, but fails to be built into cytoplasmic filaments (Supplementary Fig. 2b). Furthermore, expression of fluorophore-tagged Act5C induces fast flows[35]. Thus, in order to study the actin mesh in living cells, we used the F-actin binding protein UTRN.GFP, ubiquitously expressed under the *sqh* promotor (Fig. 2c, c′)[36]. UTRN.GFP consists of the calponin homology domain of human Utrophin fused to GFP, strongly binding to F-actin, but not actin monomers[37]. Based on fixed samples, UTRN.GFP has no effect on the morphology of the F-actin network, or on the timing of its formation and disappearance (Supplementary Fig. 2c). Thus, UTRN.GFP constitutes a suitable probe to visualize cytoplasmic F-actin in living oocytes.

Movies of UTRN.GFP revealed that cytoplasmic F-actin is highly motile and seems to "flow" randomly through the ooplasm (Supplementary Movie 1). Those movies also suggest that the actin "mesh" does not constitute an interconnected stable meshwork, but rather resembles a network of constantly assembling and disassembling filaments, that may intertwine when in close proximity. Attempts to quantify the dynamic behavior of the actin network by particle tracking or PIV were unsuccessful, mainly because of the large level of noise and crowding of the structure. We thus combined DDM with confocal imaging (Con-DDM)[38] to assess its motility. Notably, the same advection–diffusion model used for interpreting vesicle motion describes accurately the motion of cytoplasmic F-actin (Fig. 2d). Fitting of the *q*-dependent relaxation rates $\Gamma_1(q) = v_{act}q$, and $\Gamma_2(q) = D_{act}q^2$ provides an estimate for the characteristic large-scale velocity $v_{act} = 36 \pm 15$ nm s⁻¹ and for the effective diffusivity $D_{act} = (6 \pm 1.5) \times 10^{-3}$ µm² s⁻¹ of the actin filaments (where "act" stands for actin, Fig. 2e, Table 1). Such effective diffusion encompasses the actual diffusion of the center of mass of actin filaments and any active processes that reshape the actin network.

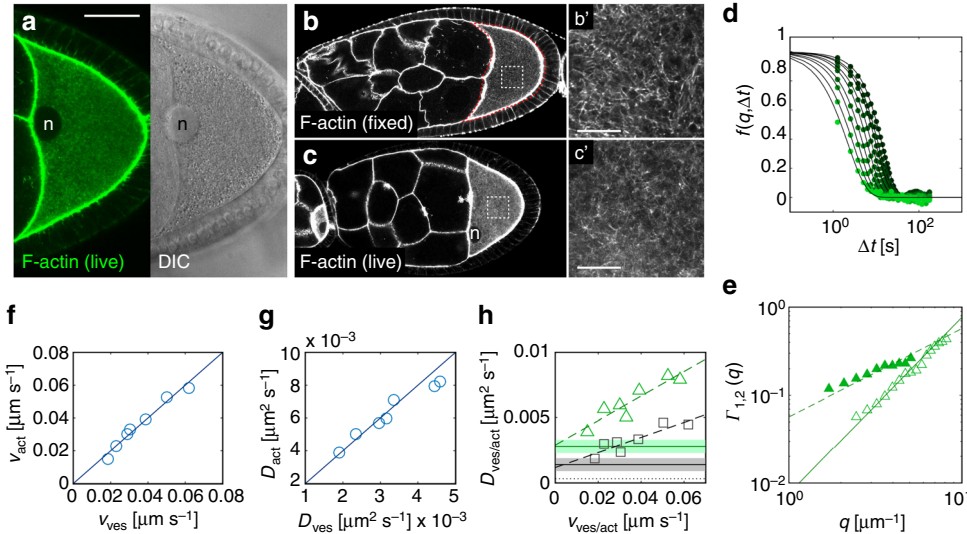

**Fig. 2** The motion of cytoplasmic F-actin directly correlates with the motion of vesicles. **a** Confocal image of F-actin stained by UTRN.GFP (left panel). DIC image of the same oocyte (right panel). n nucleus. Scale bar is 30 μm. **b** Distribution of F-actin (stained with TRITC-phalloidin) in a fixed st9 egg chamber. **b**′ High magnification of cytoplasmic F-actin (white box in **b**). **c** UTRN.GFP-expressing living egg chamber. UTRN.GFP labels the same structures as phalloidin in fixed samples (compare to **b**). **c**′ High magnification of cytoplasmic F-actin in a UTRN.GFP-expressing living oocyte (white box in **c**). **b**′, **c**′ Scale bars represent 10 μm. **d** Intermediate scattering functions (ISF) $f(q,\Delta t)$ obtained from Con-DDM analysis for different wave vectors $q$ in the range 2 μm$^{-1}$ < $q$ < 8 μm$^{-1}$. Continuous lines are best fit to Eq. (1). **e** Decorrelation rates $\Gamma_1(q)$ (solid triangles) and $\Gamma_2(q)$ (open triangles) obtained from the fit of the ISF, represented in **d**. Dashed line constitutes the best fit of $\Gamma_1(q)$ to a linear function $\Gamma_1(q) = v_{act}q$. Continuous line is obtained from the fit of $\Gamma_2(q)$ to a quadratic function $\Gamma_2(q) = D_{act}q^2$. **f** F-actin mean speeds ($v_{act}$) plotted against vesicle mean speeds ($v_{ves}$) with each data point corresponding to one cell. The continuous line represents $v_{act} = v_{ves}$. **g** F-actin diffusion coefficients ($D_{act}$) plotted against vesicles diffusion coefficients ($D_{ves}$). Each data point corresponding to one cell. The continuous line corresponds to $D_{act} = 2D_{ves}$. **h** $D_{act}$ (green triangles) and $D_{ves}$ (black boxes) as a function of the respective mean speeds $v_{act}$ and $v_{ves}$. While $v_{act}$ and $v_{ves}$ are similar for each cell, $D_{act}$ is consistently higher (~twofold) compared to $D_{ves}$. Horizontal solid lines represent $D_{act,nf}$ and $D_{ves,nf}$, obtained from colchicine-treated cells, showing no persistent motion (green—F-actin, black—vesicles, also compare to Fig. 3). Dashed areas correspond to mean value ± sd. These values agree remarkably well with the extrapolated behavior for $v \rightarrow 0$ of the experimental data obtained from control cells (dashed lines). The horizontal dotted line corresponds to the estimated value of the thermal diffusion coefficient $D_{TH}$ of the vesicles, characterizing their spontaneous fluctuation in the absence of any active process

The quantities $v_{ves}$, $D_{ves}$, $v_{act}$, $D_{act}$, display some variability from cell to cell, but exhibit interesting correlations. Comparing the results of F-actin with those on DIC vesicles revealed that the large-scale velocity $v_{act}$ in each cell compares very well with the vesicle velocity $v_{ves}$ (Fig. 2f), as $v_{act} \cong v_{ves}$. This result indicates that both F-actin and vesicles move in a persistent manner by advection, most likely driven by flows. In addition, we found a remarkable correlation between the diffusion coefficient of DIC vesicles and the effective diffusivity of F-actin, as $D_{act} \cong 2D_{ves}$ (Fig. 2g). This correlation suggests the existence of an important dynamic link between the motion of both vesicles and cytoplasmic F-actin. This hypothesis is compatible with data from mouse oocytes, where cytoplasmic F-actin seems to polymerize from the surface of vesicles[16,39]. However, a comprehensive comparison of all motions displayed by vesicles and the actin network was not performed in these studies.

**Microtubule-dependent flow enhances active diffusion.** Our results suggest that the diffusive motion of DIC vesicles and F-actin results from the combination of an intrinsic component and a flow-dependent contribution, which is evident when plotting the diffusion coefficient $D$ of either the vesicles or actin as a function of the corresponding velocity $v$ (Fig. 2h). The linear dependence of $D_{ves}$ on $v_{ves}$ is well captured by the fitting function $D_{ves} = D_{ves,0}(1 + a_{ves}v_{ves})$ with $D_{ves,0} = (1.2 \pm 0.5) \times 10^{-3}$ μm$^2$ s$^{-1}$ and $a_{ves} = (2 \pm 1) \times 10^{-2}$ s nm$^{-1}$, where $D_{ves,0}$ is the diffusion coefficient for the intrinsic component and $1/a_{ves} = 50$ nm s$^{-1}$, the typical velocity above which flows considerably affect diffusion.

Notably, the flow-independent diffusion coefficient $D_{ves,0}$ is significantly larger than the thermal diffusion coefficient $D_{TH} = 3.1 \times 10^{-4}$ μm$^2$ s$^{-1}$ (TH for thermal) that can be estimated for DIC vesicles (average size $1 \pm 0.2$ μm) and based on previous measurement of the viscosity ($\eta = 1.4$ Pa s) for these oocytes (horizontal dotted line in Fig. 2h)[14]. We can thus conclude that $D_{ves,0}$ describes an active diffusion process, devoid of any flow contributions. Similar results were found for actin, since the effective diffusivity $D_{act} = D_{act,0}(1 + a_{act}v_{act})$ is made of an intrinsic and a flow-dependent contribution, with $D_{act,0} = (3 \pm 1) \times 10^{-3}$ μm$^2$ s$^{-1}$ and $a_{act} = (3 \pm 1) \times 10^{-2}$ s nm$^{-1}$ (Fig. 2h). This suggests that both DIC vesicles and cytoplasmic F-actin display an active diffusive motion, with a component that depends on cytoplasmic streaming, and a component that does not.

To further assess the influence of flows on cytoplasmic dynamics we first attempted to eliminate kinesin-1[13,24], but the morphology of the actin mesh is disturbed in oocytes lacking kinesin (Supplementary Fig. 3a, b)[40]. However, it is known that the mesh is present in oocytes without microtubules[15], a fact that we used to test our hypothesis in oocytes obtained from females fed with the microtubule depolymerizing drug colchicine (Fig. 3a, b and Supplementary Fig. 3c, d). In oocytes without microtubules—and consequently without flows—the ballistic movement of cytoplasmic F-actin and DIC vesicles was completely abrogated, while a random diffusive motion was still detected (Fig. 3c and Supplementary Movie 2).

In fixed samples, we did not detect major changes in F-actin levels or organization in colchicine-treated oocytes. However, the actin mesh in living colchicine-treated cells looks different to

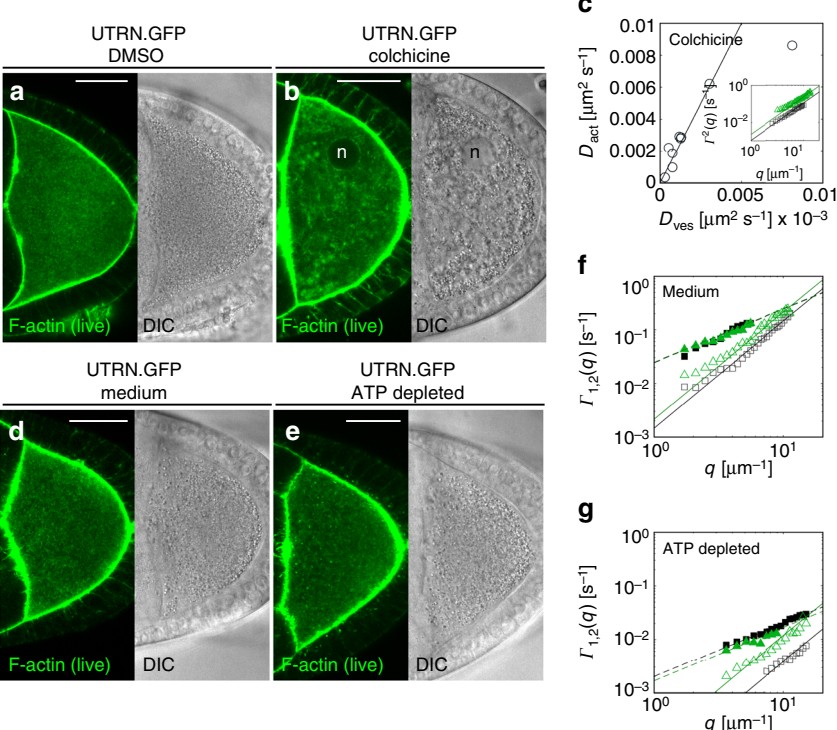

**Fig. 3** Microtubules and ATP are required for advection and active diffusion. **a, b** Persistent motion enhances active diffusion and depends on microtubules. UTRN.GFP (green, left panel) and DIC (right panel) images from living egg chambers, obtained from females fed with DMSO (**a**) or colchicine (**b**). The efficiency of colchicine feeding was assured by using only cells that displayed a nucleus (n) detached from the anterior membrane (compared to Fig. 2a). Depolymerization of microtubules, and consequently the absence of flows, causes a heterogeneous distribution of vesicles. At the same time, prominent actin filaments can be observed traversing the cytoplasm (see also Supplementary Movie 2). **c** $D_{act}$ as a function of the diffusion $D_{ves}$. In the absence of persistent flows the diffusivity of both, F-actin and vesicles, is reduced. However, the diffusion coefficients still maintain a linear relation with each other (solid line corresponds to $D_{act} = 2D_{ves}$). Each data point represents one cell. The inset shows the diffusive relaxation rates $\Gamma_2(q)$ measured for F-actin (green triangles) and vesicle (black boxes) in a representative experiment on a single cell. Continuous lines are best fits to the data with a quadratic function. **d, e** Cytoplasmic motility is driven by active processes. UTRN.GFP (green, left panel) and DIC (right panel) images from living egg chambers treated with control medium (**d**) or sodium azide and 2-D-deoxyglucose to deplete ATP (**e**, see also Supplementary Movie 4)). **f** $\Gamma_1(q)$ (empty symbols) and $\Gamma_2(q)$ (solid symbols) measured in a representative control cell (treated with medium). Green triangles correspond to F-actin, while gray squares correspond to vesicles. Continuous and dashed lines are best fits of the data to a quadratic or a linear function, respectively. **g** $\Gamma_1(q)$ and $\Gamma_2(q)$ obtained from ATP-depleted cells plotted against the wave vector $q$. Color code as in **f**. Compared to controls, the dynamics of ATP-depleted cells is more than one order of magnitude slower. Scale bar is 30 μm

control cells, with filaments being more prominent (Supplementary Movie 2). Thus, the reduction in diffusion observed in oocytes without microtubules could in principle result from changes in the mesh. However, the values found in the absence of flows for $D_{act,nf} = (2.8 \pm 1) \times 10^{-3}$ μm² s⁻¹ and $D_{ves,nf} = (1.4 \pm 0.5) \times 10^{-3}$ μm² s⁻¹ (nf = "no flow") are in excellent agreement with the values $D_{ves,0}$ and $D_{act,0}$ obtained as extrapolation to zero-flow conditions of the data in control cells, and thereby the presence of flows (Fig. 2h, Table 1). Together, these results suggest that microtubule-dependent flows significantly contribute to active diffusion. Since the females are fed colchicine for 16 h, the ovaries could in theory harbor defects in the transport of material from nurse cells (NCs) to the oocyte. We repeated our analysis with cells treated with colchicine ex vivo for only 5 min. Those cells displayed motion defects indistinguishable from cells obtained from colchicine-fed females. Therefore, we conclude that potential defects in NCs-to-oocyte transport are unlikely to influence our results.

A final interesting correlation, valid also when microtubules are depolymerized, is that actin diffusivity is always higher than vesicle diffusivity ($D_{act} \cong 2D_{ves}$) (Fig. 2g, h and Fig. 3c). The active diffusion of vesicles is seemingly locked to the actin mesh, suggesting that active vesicle diffusion might originate from an intrinsic activity of the F-actin network. This picture is in qualitative agreement with a recent model[41], describing the motion of probe particles caged in the cytoskeleton, the latter acting as an actively rearranging harmonic trap for the former. Unfortunately, the rearrangement dynamics of the F-actin network in this model was not explicitly accounted for, as its effect on a tracer particle is described as an effective random force. Moreover, a key assumption of this model is that the typical energy provided by the non-equilibrium active network to the particle does not depend on the particle properties. While this is a reasonable assumption for a passive probe particle[41], the same is less obvious for a cellular component (like a vesicle) that is known to have specific interactions with its environment and with the cytoskeleton in particular.

Of note, our DIC imaging does not reveal all the vesicles, whose size is widely distributed[42]. To test whether other organelles display different motilities, we analyzed the motion of YFP-tagged mitochondria by Con-DDM (Supplementary Movie 3). At mid-oogenesis, mitochondria display an average size of $0.82 \times 0.2$ μm[43], and smaller diffusion coefficients and velocities than DIC imaged vesicles (Table 1). This finding indicates that not all organelles display the same motility and might reflect specific biochemical mechanisms not yet discovered.

Nevertheless, the observed correlation between DIC vesicles and F-actin motility is compatible with data from mouse oocytes, in which vesicles are transported by myosin walking along a F-actin mesh, whose nodes are also occupied by vesicles[16,39]. An important difference between mammalian and *Drosophila* oocytes is that at the time the actin mesh is present, microtubules are engaged in forming the meiotic spindle in mammals, while in *Drosophila* oocytes, microtubules display an organization similar to a cell in interphase, active in transport. Despite these differences, our results suggest that the dynamic link between vesicles and actin, and the molecular mechanism responsible for it, are conserved, and survive the presence of microtubule-dependent flows in *Drosophila*. This is further supported by the fact that the murine actin mesh—like the *Drosophila* one—depends on the activity of the nucleators Spire and Formin2[44].

**Cytoplasmic motility is driven by active processes**. ATP constitutes the major cellular energy source, and ATP-dependent random force fluctuations are observed in prokaryotes and eukaryotes[8,10,11]. The pattern of motility in our cells prompted us to test the active nature of the diffusion-like motion. We depleted ATP by treating ovaries with sodium azide and 2-D-deoxyglucose shortly before imaging. After acute treatment, the actin mesh remains intact in some cells, and ATP depletion leads to an immediate reduction in all motion: persistent and diffusive behavior of vesicles and actin, are substantially abrogated (Fig. 3d, e and Supplementary Fig. 3f–h and Supplementary Movie 4).

A reliable quantification of the residual dynamics by DDM was difficult, due to presence of a marked plastic deformation of the cell. Nevertheless, in those cells in which deformation was less pronounced, the dynamics of both vesicles and actin could be successfully measured by using Eq. (1), where now the advection velocity coincides with the drift velocity. In ATP-depleted cells, we find a residual diffusive-like motion, existing on top of the constant drift (Fig. 3f). The average values of the diffusivities associated with vesicles and actin motion are $D_{ves,ATPd} = (1 \pm 0.5) \times 10^{-4} \, \mu m^2 \, s^{-1}$ and $D_{act,ATPd} = (1.4 \pm 0.8) \times 10^{-4} \, \mu m^2 \, s^{-1}$, respectively (ATPd stands for "ATP depleted"). These values are about 30 times smaller than the ones obtained in controls (Table 1).

We stress that since this is an acute treatment, the development of the egg chamber is normal up to the treatment. Similar to our ex vivo colchicine treatment, ex vivo ATP depletion affects motion within minutes of treatment, supporting further our understanding that possible defects in other areas of the egg chamber (such as motion within the NCs, or NCs-to-oocyte transport) do not have major impacts on our findings within the oocyte.

Thus, ballistic and diffusive motions displayed by vesicles and cytoplasmic F-actin are strongly ATP dependent. Since the persistent motion relies entirely on microtubule-based processes (transport and advection), we then asked what the source of the active diffusion is.

**Cytoplasmic F-actin is the major source of active diffusion**. F-actin is the source of active diffusion processes in various cell types[8,29–31]. We decided to investigate whether the cytoplasmic actin network might be driving the diffusive behavior of vesicles in the oocyte. We first tested whether active force fluctuations depend on the level of cytoplasmic F-actin. For this purpose, we studied motion of F-actin and DIC vesicles in oocytes overexpressing Spire. SpireB is the only one of four isoforms able to rescue all aspects of *spire* mutants[32].

Nevertheless, the effect of SpireB overexpression in wild-type cells has never been studied.

Driving SpireB expression under the control of the germline specific driver *nanos*-Gal4 (*nos*-Gal4) results in stratification of the ooplasm and female sterility. In addition, SpireB-overexpressing oocytes lack any detectable st9 streaming (Supplementary Movie 5, Table 1) and the dynamics are thus substantially diffusive-like, with diffusivities $D_{ves,SpireOE} = (1.4 \pm 0.5) \times 10^{-3} \, \mu m^2 \, s^{-1}$ and $D_{act,SpireOE} = (2.3 \pm 0.8) \times 10^{-3} \, \mu m^2 \, s^{-1}$. These values are decreased when compared to control cells (*nos*-Gal4 > RFP), and are similar to the ones obtained when microtubules are depolymerized, and flows are abrogated (Table 1). However, in the present case we were unable to reliably study the connection between the dynamics and the spatial correlation properties of actin and vesicles, since SpireB overexpression causes a marked cell-to-cell variability in vesicle size that prevents a meaningful sizing with the available data.

SpireB overexpression also induces the formation of F-actin in the cytoplasm of NCs (Fig. 4a, b and Supplementary Fig. 4a)[32]. To verify that F-actin concentrations are also higher in the oocyte, we stained for F-actin in fixed cells, using TRITC-labeled phalloidin, and measured the mean fluorescence intensities. The fluorescence intensities in SpireB-overexpressing cells are increased by 1.5-fold compared to controls, showing that the overall amount of F-actin is increased (Supplementary Fig. 4b). We also measured the mean distance between bright actin filaments as an indicator of the mesh size and found no apparent difference with controls.

The overexpression of Capuccino increases the density of the actin network (Supplementary Fig. 4a, b and ref. 45), and affects motion in a similar way to the overexpression of SpireB (Table 1). Although this finding seems to contradict recent results[45], temporal resolution, the method for analyzing motion and the level of detail in the analysis are different between the two studies.

Our data confirm that both flows and active diffusion depend on a well-regulated concentration of cytoplasmic F-actin, and further support a close link between the motion of cytoplasmic F-actin and vesicles. In this regard, we point out that features in motion can also be used as a means to identify candidate factors regulating the cytoskeleton. For example, vesicles in oocytes lacking Arpc1, a subunit of the Arp2/3 actin nucleation complex[46] display reduced diffusive and ballistic motion (Table 1), suggesting a role of the Arp2/3 complex in the dynamics of the cytoskeleton in the oocyte. Our results indicate that the Arp2/3 complex is not essential for the formation of the actin mesh. However, the morphology of this network is aberrant in Arpc1 mutant oocytes (Supplementary Fig. 4c, d), which might explain the observed drop in motion. The details of the involvement of the Arp2/3 complex in motion and mesh dynamics need to be further investigated.

Cross-linking between microtubules and actin filaments has been suggested as a major mechanism to regulate cytoskeletal organization. In addition, the spectraplakin-type actin/microtubule cross-linker Short stop (Shot) is involved in the polarization of the oocyte and anchoring of microtubules to the cortex[25]. Since st9 oocytes cannot be obtained from *shot* null mutants[47], we analyzed motion in oocytes expressing *shot*.RNAi. Oocytes with reduced *shot* levels display an aberrant distribution of F-actin, a general drop in mesh densities and increased diffusivity of vesicles compared to controls ($D_{ves,shot.RNAi} = (4.6 \pm 2) \times 10^{-3} \, \mu m^2 \, s^{-1}$ vs. $D_{ves,control} = (2 \pm 0.5) \times 10^{-3} \, \mu m^2 \, s^{-1}$) (Supplementary Fig. 4e, f and Table 1). These findings were confirmed in oocytes harboring a mutant allele of *shot* (*shot*[kakP1]), which abolishes the generation of isoforms that contain an actin binding domain ($D_{ves,kakP1} = (4 \pm 1) \times 10^{-3} \, \mu m^2 \, s^{-1}$ vs. $D_{ves,control} = (3 \pm 2) \times 10^{-3} \, \mu m^2 \, s^{-1}$)[47]. Even though these observations need further investigation, our

DDM analysis suggests that cross-linking between microtubules and actin is involved in the diffusive motion of vesicles.

Finally, we asked how the motion of DIC vesicles changes in oocytes lacking cytoplasmic F-actin completely. *spire* mutant oocytes do not show any obvious defects in cortical actin, but lack the cytoplasmic F-actin network and display premature fast streaming (Fig. 4c, d)[15,24,35,48]. DIC–DDM analysis revealed flow velocities in *spire* mutant oocytes to be nearly 5-fold faster than controls ($v_{ves} = 150 \pm 70$ nm s$^{-1}$ and ref. [32]), making the detection of any superimposed diffusive motion difficult (Table 1 and Supplementary Movie 5). To yet be able to measure diffusivity, we eliminated flows by treating *spire* mutant oocytes with colchicine. In oocytes lacking cytoplasmic F-actin and microtubules we measured a residual, slow, diffusive-like vesicle motion, with an effective diffusion coefficient $D_{ves,spire} = (6 \pm 2) \times 10^{-4}$ µm$^2$ s$^{-1}$ (Table 1, Supplementary Movie 5). This value is five times smaller than the value in controls and is twice the thermal diffusion coefficient $D_{TH} = 3.1 \times 10^{-4}$ µm$^2$ s$^{-1}$, estimated with the viscosity value obtained in ref. [14]. The diffusivity $D_{ves,spire}$ obtained for colchicine-treated *spire* mutants is six times larger than the diffusivity $D_{ves,ATPd} = (1 \pm 0.5) \times 10^{-4}$ µm$^2$ s$^{-1}$ from ATP-depleted cells. This difference might either be due to the presence of microtubule and F-actin independent residual active processes, or by changes in viscosity due to the lack of both microtubules and F-actin.

## Discussion

In the *Drosophila* oocyte, streaming promotes the mixing and transport of cytoplasmic components[13,49] and is required for essential events for embryogenesis, such as the localization of mitochondria[50] and developmental determinants[5,51]. Not surprisingly, streaming is more efficient than thermal diffusion in transporting large organelles over long distances[52]. However, recent studies in cells, which probably lack microtubule-dependent streaming, outlined the important role of active diffusion mechanisms, that at least in mouse oocytes are seemingly dependent on cytoplasmic F-actin[10,29,30,53]. It is thus worth assessing whether diffusion still plays a role in the presence of streaming and, if that is the case, characterizing the interplay between directed and random motion. Here we have developed a robust methodology allowing, for the first time, to separate flow induced persistent motion from the random active diffusion inside cells. We also explored how cytoplasmic F-actin affects particle dynamics, a question that is largely unknown.

Compared to existing techniques, like particle tracking or PIV, the strength of our approach lies in the ability to quantitatively discriminate between diffusive and persistent motions, without tracking (Fig. 5). This enables us to analyze the dynamic behavior of complex, intracellular structures, like the F-actin mesh, which was not possible before. We found that both vesicles and cytoplasmic F-actin move in a persistent manner by advection, mainly as a result of flows. In addition, they display a non-thermal, active diffusion that is dependent on cytoplasmic, but not cortical, F-actin. In fact, we speculate that cytoplasmic F-actin is the major driving force behind active diffusion in oocytes. However, we also found that active diffusion is reduced in oocytes without microtubules, a novel finding suggesting that microtubules are not only essential for transport and cytoplasmic streaming, but also substantially contribute to active diffusion. This may constitute a difference to murine oocytes, as well as various cultured cells where active fluctuations have been measured[8,31,54]. However, the involvement of microtubules in cytoplasmic motion was not investigated in detail in these studies, and thus this novel function for microtubules modulating diffusion might be conserved in other systems.

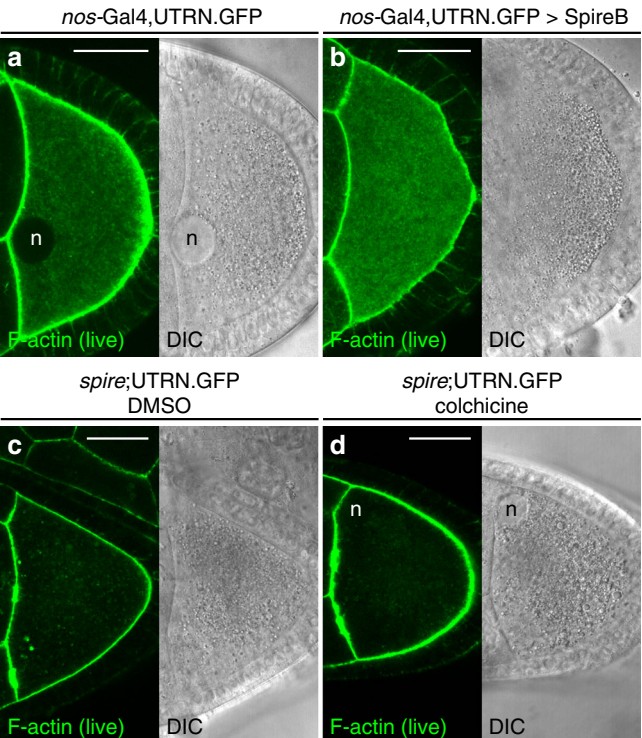

*nos*-Gal4,UTRN.GFP *nos*-Gal4,UTRN.GFP > SpireB

*spire*;UTRN.GFP *spire*;UTRN.GFP
DMSO colchicine

**Fig. 4** Cytoplasmic F-actin is the major source of active diffusion. **a** Living control egg chambers or **b** egg chambers over-expressing the actin nucleator SpireB. Overexpression of SpireB causes an increase in the amount of F-actin in the oocyte (see also Supplementary Fig. 4). **c, d** *spire* mutant oocytes were incubated in control DMSO-containing medium (**c**) or in colchicine-containing medium **d**. As expected, *spire* mutants do not form an actin mesh. Scale bar is 30 µm

Our study sheds light on the link between the motion of different cytoplasmic components, in particular between large vesicles and cytoplasmic F-actin, which are found to exhibit the same advection–diffusion dynamics. This link is robust upon perturbation of flows, and our findings suggest that cytoplasmic F-actin, a dense non-equilibrium fluctuating mesh, is the source of active diffusion for vesicles. Even though previous cell biological and biophysical observations pointed out that vesicles and actin networks were functionally linked in other cells, our work is the first one to measure and compare the diffusive-like motion of both vesicles and the F-actin network, finding a quantitative relation between the parameters describing the dynamics of both structures. In the future, this link will need to be studied in more depth, for instance by performing a detailed analysis of the correlation between the size of the actin mesh and of the vesicles, as a function of the vesicle size.

## Methods

**Fly stocks and genetics**. If not stated otherwise, all flies where kept on standard cornmeal agar at room temperature. Fly stocks used in this study where: w[1118], w; CyO/Sco;P{sqh-UTRN.GFP}/TM6B,Hu[1],Tb[1][37], w;CyO/Bl;P{UASp-SpirB.tdTomato} and w;CyO/Bl;P{UASp-Capu.tdTomato} (from M. Quinlan), w;FRTG13,shot[kakP1] (from K. Röper), w;;P{GAL4::VP16-nos.UTR} (BL4937), w;FRTG13,Khc[27] \CyO[55], w;spir[1]/CyO;P{sqh-UTRN.GFP}/TM6B,Hu[1],Tb[1], w;Df(2 L)Exel6046/ CyO (BL7528), w;P{UAS-RFP.W}2 (BL30556) w[*];P{UASp-GFP.Act5C}3 (BL9257), w[*];P{UASp-Act5C.mRFP}38 (BL24779), Arpc1[Q25sd] (BL9137), sqh-mito.YFP (BL7194), shot.RNAi[GL01286] (BL41858) and Jupiter.GFP[56]. Except otherwise stated, all UAS/Gal4 crosses were performed at 25 °C, germline clones were induced using the FLP;FRT *ovoD* system. The stock w;CyO/Sco;nos-Gal4. VP16,sqh-UTRN.GFP/TM6B was generated by the recombination of the respective chromosome (this study).

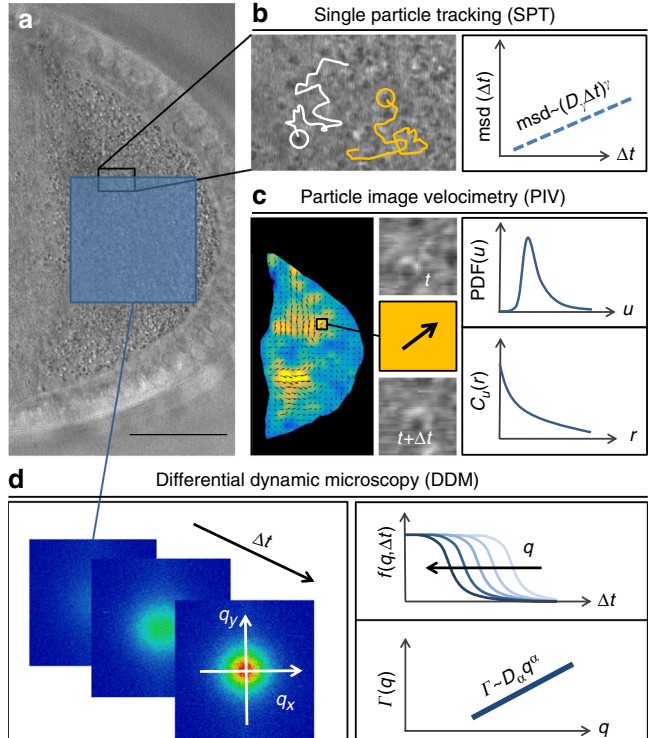

**Fig. 5** Experimental approaches to quantify intracellular motion. **a** Representative st9 oocyte imaged by DIC microscopy. Scale bar is 30 μm. **b** SPT relies on the identification of specific particles in each time frame, and linking their position across different frames to reconstruct a trajectory (left panel). The typical output of SPT analysis is the mean squared displacement (MSD). The exponent γ, characterizing the dependence of the MSD on the delay time Δt in a given regime, indicates the nature of motion (e.g., diffusive, superdiffusive), while the prefactor $D_\gamma$ represents a generalized diffusion coefficient. **c** PIV enables the reconstruction of a coarse-grained velocity map, where a 2D vector **u**, representing the local velocity, is associated to each point on a regular grid (left panel). **u** is determined by estimating the average net displacement across two time frames (δt) of the particles comprised within a small window (middle panels). PIV analysis allows the characterization of the statistical properties of flows associated with the ballistic motion of the tracer particles, for example the probability distribution function PDF(u) of speeds or the spatial velocity correlation function $C_u(r)$ (right panel). **d** Compared to SPT and PIV, DDM is not based on direct-space analysis. Instead, DDM quantifies the dynamics in the Fourier space, measuring the relaxation of spontaneous density fluctuations occurring at different wave vectors q (left panel). This relaxation process is captured by the intermediate scattering functions f(q,Δt), represented on semi-logarithmic scales for various values of q. The behavior as a function q of the characteristic relaxation rate Γ associated with the decay of f(q,Δt) provides a key information about the nature of the motion and allows to determine the relevant parameters. The lower left panel represents the simple and important case where Γ has a power-law dependence on q (log–log scale). This occurs, for example, when the underlying dynamic is diffusive (α = 2 and $D_\alpha$ is the diffusivity) or ballistic (α = 1 and $D_\alpha$ coincides with a characteristic speed)

**Germline specific *short stop* knock down.** Female flies of the genotype *nos*-Gal4. VP16,*sqh*-UTRN.GFP were crossed to *shot*.RNAi[GL01286] males and progeny was raised at 25 °C. At this temperature, reduction of shot causes cell death and no egg chambers could be recovered. Therefore, freshly hatched females were kept at 18 °C for 5 days, allowing the formation of stage 9 oocytes. Before dissection, females were kept at 25 °C for one day and subsequently fattened with dry yeast at 25 °C over night. Egg chambers were dissected and movies acquired as stated below.

**Actin mesh staining in fixed samples.** Ovaries where dissected and fixed in 10% methanol free formaldehyde in PBS, containing 0.1% Tween-20 (PBT0.1), for a maximum of 10 min. The fixed cells were washed 4× in PBT0.1 and stained with 1 μM TRITC-coupled phalloidin (in PBT0.1, Sigma-Aldrich) over night at 4 °C. The stained samples were washed 4× in PBT0.1 and mounted in Vectashield mounting medium (Vectorlabs). Images were acquired on a Leica SP5 inverted confocal microscope, using a 40×/1.3 Oil DIC Plan-Neofluar or a 100×/1.4 Oil DIC objective. Images were taken within 24 h after staining.

**Live imaging of the actin mesh.** Ovaries were dissected in a drop of halocarbon oil (Voltalef 10S, VWR) on a glass coverslip and single egg chambers were separated using fine tungsten needles. Images were acquired on a Leica SP5 inverted confocal microscope, using a 40×/1.3 Oil DIC Plan-Neofluar or a 100×/1.4 Oil DIC objective.

For high-resolution image series, a single plane from the middle of the oocyte was imaged at a scan speed of 100 Hz and an image resolution of 1,024 × 1,024 pixels (corresponding to one image every 10.4 s).

For DDM analyses images where taken at a scan speed of 400 Hz and a resolution of 1,024 × 512 pixels, corresponding to one image every 1.29 s. The pinhole diameter was set to a corresponding thickness of the image plane of about 1.39 μm. The cells were illuminated by 488 nm and 561 nm laser light and emission light was collected simultaneously using a hybrid detector at 500–550 nm (GFP), a conventional photon multiplier at 560–650 nm (vesicle auto-fluorescence) and a transmitted light detector with DIC filter set.

**Drug treatment.** Depolymerization of microtubules was achieved by either feeding (Fig. 3) or treatment of dissected egg chambers (Fig. 4). For the feeding experiment, 200 μg ml$^{-1}$ colchicine were diluted in yeast paste and fed to female flies for 16 h at 25 °C. For short-term treatment, flies were fattened overnight, ovaries dissected in dissection medium (1× Schneider's medium +2% DMSO) and treated in 20 μg ml$^{-1}$ colchicine in dissection medium for 5 min at room temperature (Fig. 4). Ovaries were washed once in a drop of dissection medium and dissected in halocarbon oil. Imaging was performed as described above. The depolymerization of microtubules in the oocyte by these treatments was confirmed by feeding flies and treating oocytes expressing the microtubule binding protein Jupiter.GFP, as well as by scoring those oocytes that displayed a misplaced nucleus, a robust read out of microtubule depolymerization.

ATP was depleted by treating dissected ovaries in 0.4 mM NaN3 and 2 mM 2-Deoxy-D-glucose in dissection medium for 5.5 min at room temperature (Fig. 3 and Supplementary Fig. 3f–h). Ovaries were washed in a drop of dissection medium and further dissected and imaged in a drop of halocarbon oil.

**Differential dynamic microscopy.** DDM analysis was performed by using both DIC imaging (DIC–DDM) and confocal imaging (Con-DDM). While Con-DDM was previously demonstrated with densely packed bacteria and colloids[38], both techniques are employed here for the first time to probe the cell's interior dynamics. Time-lapse movies acquired at a fixed frame rate with either imaging mode are treated in the same way. Reciprocal space information is extracted from the analysis of the N intensity frames of the video $i_n(\mathbf{x}) = i(\mathbf{x}, n\delta t)$ acquired at times $n\delta t$ (n = 1, …, N) by calculating their spatial Fourier transform $I_n(\mathbf{q}) = \int e^{j\mathbf{q}\cdot\mathbf{x}} i_n(\mathbf{x})\mathbf{d}^2 x$. Here j is the imaginary unit, δt is the inverse frame rate of the video acquisition, $\mathbf{x} = (x, y)$ are the real space coordinates and $\mathbf{q} = (q_x, q_y)$ are the reciprocal space coordinates. DDM analysis is based on calculating the image structure function defined as

$$d(\mathbf{q}, \Delta t) = \left\langle |I_{n+m}(\mathbf{q}) - I_n(\mathbf{q})|^2 \right\rangle_n,$$

where the average $\langle \ldots \rangle_n$ is made over image pairs that are separated by the same time delay $\Delta t = m\delta t$ over the entire stack. The azimuthal average

$$d(q, \Delta t) = \langle d(\mathbf{q}, \Delta t) \rangle_{q=\sqrt{q_x^2 + q_y^2}},$$

is also typically performed whenever the structure and the dynamics of the sample are isotropic and spatially homogeneous in the image plane, as in the present study. Once the image structure function is calculated, it can be fitted to the following theoretically expected behavior

$$d(q, \Delta t) = A(q)[1 - f(q, \Delta t)] + B(q)$$

Here B(q) is a background term due to the detection noise, A(q) is an amplitude term that contains information about the imaging mode used and the distribution of individual entities in the image, and f(q,Δt) is the so-called ISF, which encodes the information about the sample dynamics[17,57]. For instance, diffusive motion with diffusion coefficient D is characterized by $f(q, \Delta t) = e^{-Dq^2\Delta t}$ and ballistic motion with constant velocity $v_0$ is described by $f(q, \Delta t) = e^{-jv_0\cdot\mathbf{q}\Delta t}$[18].

A case of interest for the present study is the one of a collection of particles moving via a combination of ballistic and Brownian motion, the first being characterized by a constant velocity **v** drawn from a prescribed distribution p(**v**),

the second by a diffusion coefficient $D$. In this case the ISF reads

$$f(\mathbf{q}, \Delta t) = e^{-Dq^2\Delta t} \int p(\mathbf{v}) e^{-j\mathbf{v}\cdot\mathbf{q}\Delta t} d^3\mathbf{v},$$

If the velocity distribution is isotropic, it can be written in terms of the speed distribution $p_{s,3D}(v)$

$$p(\mathbf{v}) = \frac{1}{4\pi|\mathbf{v}|^2} p_{s,3D}(|\mathbf{v}|),$$

that we assume to be in the form of a Schulz distribution

$$p_{s,3D}(v) = \frac{v^Z}{Z!} \left(\frac{Z+1}{\bar{v}}\right) e^{-\frac{v}{\bar{v}}(Z+1)}.$$

Here $Z$ is a shape parameter (set to 2 in our case) and $\bar{v}$ is the average speed:

$$\bar{v} = \int v p_{s,3D}(v) dv.$$

In order to make explicit the dependence on the average speed in the ISF one can also consider the rescaled velocity distribution $p_0$, defined as: $p_0(e) = \bar{v}^3 p\left(\frac{e}{\bar{v}}\right)$. In terms of its Fourier transform $P_0$, the ISF can be written

$$f(\mathbf{q}, \Delta t) = e^{-Dq^2\Delta t} P_0(\Gamma_1(q)\Delta t),$$

where $\Gamma_1(q) = \bar{v}q$ is $q$-dependent decorrelation rate.

If a 2D projection of the motion is considered, the speed distribution reads

$$p_{s,2D}(v) = \int p(\mathbf{v})\delta\left(v - \sqrt{v_x + v_y}\right) d^3\mathbf{v}$$

where $\delta$ is the Dirac delta function. This is the quantity that is typically measured with PIV. It is important to note that, in general, the mean value of the 2D projected velocity field:

$$\bar{u} = \int v p_{s,2D}(v) dv,$$

does not coincide with $\bar{v}$. For example, in our case (Schulz distribution with $Z = 2$), it can be shown that $\bar{u} = 0.566\bar{v}$.

The choice of the Schulz distribution is motivated by the simplicity of its analytical form, that makes the fitting procedure of the experimental ISF particularly simple and robust, and it is supported by the good agreement with the 2D speed distribution measured with PIV (Fig. 1d–f).

The accessible $q$-range for DDM analysis is given, in principle by $[2\pi/L, \pi/a]$ where $L$ is the size of the considered ROI and $a$ is the effective pixel size (typically $L = 33\,\mu m$, $a = 0.13\,\mu m$). In practice, this range can be significantly reduced. This can be due, at low $q$, to the presence of very slow dynamics that cannot be fully captured during the finite duration of the image acquisition and, at high $q$, mainly to the loss of signal due to the microscope transfer function (that depress higher spatial frequencies in the image) and to the presence of dynamics that are too fast to be sampled with the experimental frame rate. Overall, in most of the experiments presented in this work, the effective q-range roughly corresponds to $[2, 20]\,\mu m^{-1}$. In the direct space, this corresponds to considering density fluctuations on length scales comprised approximately between 0.3 and 3 $\mu$m. It is worth noting that, when flow is present inside the oocyte, the velocity field exhibits some spatial heterogeneity, especially close to the cell edges. For this reason, we considered only ROIs far from these edges, which we imaged long enough to guarantee that the average velocity was rather homogeneous across the field of view.

For a given cell, three ROIs are typically considered: a larger one ($L \cong 32\,\mu m$) in the anterior part of the cell and two smaller ones ($L \cong 16\,\mu m$) placed almost symmetrically in the posterior region of the cell (Fig. 1b). Unless explicitly otherwise stated, all the reported results are obtained by averaging the image structure function obtained in the three ROIs.

**Particle image velocimetry.** Maps of the instantaneous intracellular velocities were obtained by analyzing time-lapse DIC and reflection confocal microscopy image sequences with a custom PIV software written in MATLAB[58]. The time interval between consecutive frames considered for the analysis was 2.6 s. The interrogation window was $32 \times 32$ pixels (pixel size comprised between 0.101 $\mu$m and 0.145 $\mu$m), with an overlap of 50% between adjacent windows. Only the region inside the cell perimeter was considered. Speed probability distribution functions PSD($u$) for a single cell were obtained as the normalized histogram of the speeds measured on all grid points and in all frames. Typically, about $1.2 \cdot 10^5$ vectors contribute to the speed statistics for a single cell.

**Data availability.** All relevant data and computer codes are available from the corresponding authors upon request.

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

## Acknowledgements

We thank Dr. Margot E. Quinlan for comments and reagents, and M. Wayland for assistance with imaging. We also thank M. Cosentino Lagomarsino and M. Gherardi for critical reading of the manuscript. M.D. and I.M.P. were supported by the BBSRC, the Department of Zoology (Cambridge), the University of Cambridge, and an Isaac Newton Trust fellowship to MD. F.G. and R.C. acknowledge funding by the Italian Ministry of Education and Research, Futuro in Ricerca Project ANISOFT (RBFR125H0M) and by Fondazione CARIPLO-Regione Lombardia Project Light for Life (2016-0998).

## Author contributions

M.D., F.G., I.M.P. and R.C. designed experiments, analyzed the data, discussed results and wrote the manuscript. M.D. and F.G. performed experiments.

## Additional information

**Competing interests:** The authors declare no competing financial interests.

