## [Peer Review File · Nature Communications]

Reviewers' comments:

Reviewer #1 (Remarks to the Author):

In this manuscript, Drechsler et al. combined Differential Dynamic microscopy (DDM) with either DIC or confocal fluorescence imaging to quantify the motion of vesicles and actin mesh in stage 9 *Drosophila* oocytes. By using this method, they were able to detect both the ballistic movement and the diffusive motion of vesicles as well as actin mesh. In combination with genetic mutations and chemical treatment, they showed that the ballistic movement is mostly driven by microtubule-dependent cytoplasmic flow, while active diffusion is mainly dependent on actin but also enhanced by microtubules. Overall, I think it is a solid study with unbiased quantitative methods that can be applied to cytoplasmic movement studies in other cell types without any fluorescent probe labelling.

Here are my main concerns regarding their conclusions:

1. In supplementary movie 1, it seems that there are different levels of vesicle movement within the oocyte, lower in the posterior side and higher in the anterior part, especially active near the ring canals where the transport from the nurse cells to the oocyte occurs. It has been known that the microtubule network displays an anterior-posterior gradient, and thus higher streaming in the anterior side is expected. I wonder whether the authors have ever taken this position information in consideration, and quantify the movement in different areas of the oocytes. Additionally, genetic and chemical manipulations of microtubules and actin filaments may affect nurse cell behaviours and nurse cell-to-oocyte transport. I hope the authors will at least discuss this possibility when they draw their conclusions.

2. Comparing Figure 3b to 3a and supplementary movie 2 to 1, it seems that colchicine treatment changes the actin mesh organisation, and makes more foci/filaments of actin (instead of mostly dots in control). This implies that some crosstalk between microtubules and F-actin. Thus before the authors conclude that microtubules contribute to the active diffusion, I would like the authors to address two following points: 1) In control and colchicine treatment, examine microtubules and F-actin to make sure first colchicine treatment efficiently depolymerizes microtubules, and second to detect any changes regarding the cross-linking between microtubules and F-actin. 2) *Drosophila* only has a spectraplakins gene, Shot, that crosslinks microtubules and F-actin. I hope the authors will examine the effect of shot mutant on the ballistic motion and diffusive motion of vesicles and actin mesh.

3. In the Spire overexpression experiment, it is surprising that the overexpression leads to 1.4 fold increase of F-actin but more than 10-fold decrease in flow velocity and 2-fold decrease in diffusion. As the authors suggested, it could be due to the Spire overexpression leads to hyper-stability of F-actin, and F-actin dynamics contributes to the diffusive movement. I think the authors should further test this hypothesis by treating the Spire overexpression ovaries with LatA to see whether it indeed is less sensitive to LatA treatment, and used DDM to measure the effect. It is intriguing that overexpression of Cappuccino, another actin nucleator, increases actin filament number but does not affect flow rate. The authors speculated that higher concentration of F-actin is not sufficient to stop cytoplasmic flow. However, it remains unclear whether the Capu overexpression affects active diffusion. I think the authors could test it, and furthermore, simply increase F-actin concentration by Jasplakinolide treatment to see whether the number of actin filaments affects cytoplasmic flow and active diffusion. In addition, I wonder whether Arp2/3 is involved in the actin mesh formation/dynamics, and thus active diffusion. This can be easily tested in the authors' setting with DDM when Arp2/3 is inhibited by CK-666.

4. The manuscript presented an almost perfect correlation of vesicle diffusion and actin diffusion: $D_{act} \approx 2D_{ves}$. And the authors proposed that this is due to the vesicle size is around 2 times of the actin mesh size. It is somehow surprising that the ooplasmic vesicles come in such a uniform size.

The oocyte contains lipid droplets, organelles, mRNA, and proteins that, to my understanding, may be of various diameters. Here I am curious whether the authors could test this hypothesis by examining vesicles of different sizes. For example, mitochondria undergo active fusion and fission, and display heterogeneity in size. It would be nice if the authors could perform the DDM analysis of mitochondria (by Mitotracker dye labeling, etc.) and actin mesh (by UTRN-GFP), to see whether the size of mitochondria does affect its diffusion rate.

Reviewer #2 (Remarks to the Author):

The major claim of the paper is that differential dynamic microscopy (DDM) is a potent tool to understand passive and active transport in complex environments, specifically in drosophila oocyte cells. This reviewer's experience with DDM suggests that the claim is reasonable: the method lets its user obtain dynamic information even in environments that would be difficult to explore by traditional methods (e.g., dynamic light scattering, DLS) and with better precision than most particle tracking/velocimetry algorithms (which additionally require the particles to be resolved or at least distinguishable against the background, unlike DDM).

I wish figure 1D (Γ vs q in a log-log representation) would be given one or two linear-linear insets; for the velocity motion, plot Γ vs q ; for the diffusion Γ vs q^2 . Even better might be Γ/q and Γ/q^2 vs q and q^2 respectively...should be flat lines, showing off how well the method really works.

Not being an expert in microbiology, I am unable to comment on another claim of the paper, which is that the particular cells chosen for investigation need to be studied to strengthen claims made for mammalian oocytes, but this hardly matters because this paper will advance an important tool for a wide variety of studies.

The presentation is logical, and I found only minor grammatical and mechanical flaws with the writing. I imagine the copy editors will repair these and others:

Line 39: progresses  progress

Line 76: require  requires

Line 121: contributes  contributions

Line 143: Under  From

Active diffusion and advection in the *Drosophila* ooplasm result from the interplay of the actin and microtubule cytoskeletons

Drechsler et al

We are grateful to both reviewers for the positive comments on our work and for the insightful suggestions. As explained in more detail below, we have addressed all points raised by the reviewers, and most of them have resulted in changes in the manuscript. In this letter we have used a color code to explain our actions and changes (black = reviewers comment, red = authors reply, blue = actions taken)

Reviewer 1

*In this manuscript, Drechsler et al. combined Differential Dynamic microscopy (DDM) with either DIC or confocal fluorescence imaging to quantify the motion of vesicles and actin mesh in stage 9 *Drosophila* oocytes. By using this method, they were able to detect both the ballistic movement and the diffusive motion of vesicles as well as actin mesh. In combination with genetic mutations and chemical treatment, they showed that the ballistic movement is mostly driven by microtubule-dependent cytoplasmic flow, while active diffusion is mainly dependent on actin but also enhanced by microtubules. Overall, I think it is a solid study with unbiased quantitative methods that can be applied to cytoplasmic movement studies in other cell types without any fluorescent probe labelling.*

We thank Reviewer 1 for the positive comments about our work.

Here are my main concerns regarding their conclusions:

1. In supplementary movie 1, it seems that there are different levels of vesicle movement within the oocyte, lower in the posterior side and higher in the anterior part, especially active near the ring canals where the transport from the nurse cells to the oocyte occurs. It has been known that the microtubule network displays an anterior-posterior gradient, and thus higher streaming in the anterior side is expected. I wonder whether the authors have ever taken this position information in consideration, and quantify the movement in different areas of the oocytes

In our initial submission, we presented results that were obtained by combining information from three different regions of interest (ROI) in each cell, two symmetrically positioned at the anterior end of the oocyte, and one at the posterior half of the oocyte. We chose then to combine the image structure functions from these three regions to obtain the averaged information from the whole cell. However, we agree with Reviewer 1 that stressing the space-resolving capability of DDM reinforces our message, and relates the information to known asymmetries within the oocyte.

In the revised version, we have illustrated this capability for the control cells. As anticipated by the Reviewer's comment, we did find a statistically significant difference between the dynamics in the anterior and posterior part of the cell: cytoplasmic flow is faster in the anterior region, while the effective diffusivity is larger in the posterior portion of the cell.

Nevertheless, in order to preserve both the simplicity of the message and the statistical robustness of our results, we have decided to maintain the velocity and diffusivity - averaged

over the whole cell- as the sole indicators when comparing different cell types and treatments.

a) We have now modified Figure 1b to clearly illustrate the position of the three ROIs that we have analyzed with DDM.

b) In *Results and Discussion*, we now describe the results of DDM analysis in both the anterior and posterior regions of control cells (page 6, line 184). We have still averaged the two posterior ROIs because they displayed very similar values.

.... Additionally, genetic and chemical manipulations of microtubules and actin filaments may affect nurse cell behaviours and nurse cell-to-oocyte transport. I hope the authors will at least discuss this possibility when they draw their conclusions.

We agree with the reviewer that the transport of material from nurse cells (NC) into the oocyte could influence the cytoplasmic content of the oocyte, and therefore the intracellular motion within the oocyte. However, we have several points and observations that support the model that changes in NC behavior, or NC-to-oocyte transport are not a major cause of our motion observations within the oocyte cytoplasm:

a) Egg chambers without kinesin heavy chain do not seem to display any defects in growth or NC-to-oocyte transport, suggesting that the material in the ooplasm is similar to wild type egg chambers. However, these kinesin mutant egg chambers lack cytoplasmic streaming in the oocyte, demonstrating that the lack of advection is due to the absence of the motor protein, and not to changes in the composition of the oocyte cytoplasm.

b) In the results section (page 9), we show cells derived from females that have been fed colchicine overnight (16h). Due to the putative long exposure to the drug, these oocytes could in theory harbor NC-to-oocyte transport defects. However, we repeated our analysis with egg chambers that were treated with colchicine *ex vivo* for only five minutes. Those cells displayed motion defects indistinguishable from cells obtained from colchicine fed females.

We have added this information to page 10, line 327

c) In the short-term (5 min) ATP depletion experiment, the development of the egg chamber is normal up to the treatment. Similar to our *ex vivo* colchicine treatment, *ex vivo* ATP depletion affects cytoplasmic motion within minutes of treatment, supporting further our understanding that possible defects in other areas of the egg chamber (such as motion within the NCs, or NCs-to-oocyte transport) do not have major impacts on our findings within the oocyte.

We have added this information to page 11, line 391

d) Finally, we agree that genetic manipulations of Capuccino and Spire could potentially influence other actin structures, in addition to the actin mesh within the oocyte. However, since spire mutants do not show any other actin defects, apart from the

lack of the actin mesh, we speculate that the transport from NCs to the oocyte is not affected.

Taken together, we believe that the observed defects in cytoplasmic motion are direct consequences of cytoskeletal defects within the oocyte, and not of an altered composition of the ooplasm.

2. Comparing Figure 3b to 3a and supplementary movie 2 to 1, it seems that colchicine treatment changes the actin mesh organisation, and makes more foci/filaments of actin (instead of mostly dots in control). This implies that some crosstalk between microtubules and F-actin. Thus, before the authors conclude that microtubules contribute to the active diffusion, I would like the authors to address two following points:

2.1) In control and colchicine treatment, examine microtubules and F-actin to make sure first colchicine treatment efficiently depolymerizes microtubules, and second to detect any changes regarding the cross-linking between microtubules and F-actin.

a) The depolymerization of the microtubules in the oocyte by colchicine treatment was tested repeatedly by us, as well as by others in the field. In addition, we have now run additional tests and fed colchicine to flies expressing the microtubule binding protein Jupiter.GFP. Compared to controls, no microtubules could be detected in oocytes obtained from colchicine fed flies (Figure 1 below). Of note, microtubules in the follicle cells (FC), surrounding the germline, are much less affected by the treatment. This differential effect has been previously reported several times. The mis-localization of the oocyte's nucleus (n) constitutes another read out of microtubule depolymerization, and could be observed in all cells monitored.

These read-outs for microtubule depolymerization are now described in the legend of Figure 3, and in Material and Methods (line 782).

Figure 1: Jupiter.GFP expressing egg chambers obtained from DMSO (left) or colchicine fed (right) females. Colchicine feeding lead to a complete de-polymerization of microtubules in the oocyte, but not the follicle cells. n=nucleus, FC=follicle cells.

b) The evaluation of the organization of the actin mesh in colchicine-treated oocytes was already included in the manuscript (Figure 3b and Supplementary Figure 3c, d). In our hands, monitoring the mesh in fixed colchicine-treated samples does not indicate major

changes in F-actin levels or organization. We do agree, that in living colchicine-treated cells, the actin mesh looks different to control cells, in which filaments are much more prominent.

A comment on this has now been added to page 9, line 317. In here, we have also added that "...the experimental values found in the absence of flows for $D_{act,nf} = (2.8 \pm 1) 10^{-3} \mu\text{m}^2/\text{s}$ and $D_{ves,nf} = (1.4 \pm 0.5) 10^{-3} \mu\text{m}^2/\text{s}$ (where nf stands for "no flow") are in excellent agreement with the values $D_{ves,0}$ and $D_{act,0}$ obtained as extrapolation to zero-flow conditions of the experimental data in control cells, and thereby the presence of flows (Fig. 2h, Table 1). Together, these results suggest that microtubule-dependent flows significantly contribute to active diffusion."

c) Regarding the final point, we strongly agree with the reviewer that any potential contribution of microtubule-actin crosslinking to motion within the oocyte cytoplasm is of extreme interest and should be addressed in detail in the future.

We are unaware of any technique allowing us to test microtubule-actin crosslinking directly *in vivo*, but we have investigated the function of the spectraplakins gene, *Shot*, as described below.

2.2) *Drosophila* only has a spectraplakins gene, *Shot*, that crosslinks microtubules and F-actin. I hope the authors will examine the effect of *shot* mutant on the ballistic motion and diffusive motion of vesicles and actin mesh.

We agree with the reviewer that testing *short stop* (*shot*) mutants will help to reveal a potential function for microtubule-actin crosslinking in ooplasm motion, and theoretically constitutes a straightforward experiment. However, loss of *Shot* leads to cell death in the germline¹. We have done two experiments to overcome this problem:

a) We reduced *shot* levels by expressing a specific RNAi in the germline (*nos>shot.RNAi*) and shifted the culturing temperature from 25°C to 18°C, allowing us to obtain *st.9 shot* mutant oocytes. In these *shot.RNAi* cells, we observed a series of different phenotypes, now described in page 13, Supplementary Fig. 4e-g and Table 1.

Firstly, we found that the typical topology of flows, with higher velocities anteriorly, is abolished. Compared to controls, flows in the posterior part of the *shot.RNAi* cells are now similar to the ones at the anterior end. This correlates well with the function of *Shot* in anchoring microtubules to the anterior/lateral cell cortex. It has recently been shown that *shot* mutants display an even distribution of microtubules throughout the oocyte², and therefore a more even appearance of flows was expected.

Secondly, we found that the density of the mesh seems to be reduced in both living and fixed *shot.RNAi* cells (Supplementary Fig. 4f, g), while live cells also display an accumulation of UTRN.GFP in puncta throughout the cytoplasm (Supplementary Fig. 4e). This defect is not seen in fixed cells, presumably because the structures are lost during the fixation process. Reliable values concerning the motion of F-actin could not be obtained from the mutant cells.

Thirdly, we found that diffusion coefficients are higher in *shot*.RNAi cells, while the ballistic motion seems, in average, unaffected, although displays an extreme variability. While the density of the mesh seems to be reduced in *shot*.RNAi cells, which should result in faster flows, it is known that Shot is essential for microtubules to anchor at the anterior/lateral cortical actin, and thus lack of Shot results in a dramatic disorganization of the microtubule cytoskeleton, and therefore a reduced intracellular motility could be expected.

For *shot*.RNAi we found:

$D_{ves,shot.RNAi} = (4.6 \pm 2) \times 10^{-3} \mu m^2/s$ (compared to control nanos-Ga4, UAS-RFP cells at $2 \pm 0.5 \times 10^{-3} \mu m^2/s$)

$V_{ves,shot.RNAi} = 11 \pm 7 \text{ nm/s}$ (compared to control nanos-Ga4, UAS-RFP cells at $13 \pm 3 \text{ nm/s}$)

b) We confirmed our motion data obtained from *shot*.RNAi cells in the hypomorphic allele *shot*^{kakP1}, which abolishes the expression of shot isoforms containing an actin binding domain¹.

For *shot*^{kakP1} we found:

$D_{ves,kakP1} = (4 \pm 1) \times 10^{-3} \mu m^2/s$ (compared to control cells at $3 \pm 1 \times 10^{-3} \mu m^2/s$)

$V_{ves,kakP1} = 26 \pm 26 \text{ nm/s}$ (compared to control cells at $36 \pm 15 \text{ nm/s}$)

Taken together, our result show that *shot* is crucial to maintaining the correct topology of flows (probably by regulating microtubule cortical anchoring), as well as modulating active diffusion. Since the diffusive behavior of vesicles in *shot* mutant cells does not drop (on the contrary, it increases), and is therefore different to cells in which microtubules have been depolymerized, we conclude that this observation strengthens our initial interpretation that microtubules contribute to active diffusion. However, the phenotypes we observe in *shot* mutant cells are complex and we do not know if Shot indeed cross-links microtubules and F-actin in the cytoplasm of the oocyte. Because of this, we have concluded in our manuscript that:

“...Even though these observations need further investigation, our DDM analysis suggests that cross-linking between microtubules and actin is involved in the diffusive motion of vesicles in the oocyte” (page 13 line 459”).

3. In the Spire overexpression experiment, it is surprising that the overexpression leads to 1.4 fold increase of F-actin but more than 10-fold decrease in flow velocity and 2-fold decrease in diffusion. As the authors suggested, it could be due to the Spire overexpression leads to hyper-stability of F-actin, and F-actin dynamics contributes to the diffusive movement. I think the authors should further test this hypothesis by treating the Spire overexpression ovaries with LatA to see whether it indeed is less sensitive to LatA treatment, and used DDM to measure the effect.

It is intriguing that overexpression of Cappuccino, another actin nucleator, increases actin filament number but does not affect flow rate. The authors speculated that higher concentration of F-actin is not sufficient to stop cytoplasmic flow. However, it is remains unclear whether the Capu overexpression affects active diffusion....

We think that the observed 1.5-fold increase in the mesh fluorescence intensity exhibited by SpireB overexpressing cells is difficult to be linked quantitatively to the system dynamics. In particular, we believe that the measurement of fluorescence intensities after phalloidin staining might not fully reflect the absolute levels of the actin meshwork, as intensities in LatA treated cells, where the actin mesh is destroyed, drop to roughly 1/2 of the control

value, rather than to zero (see figure 2 below). This suggests that a quantitative use of fluorescent intensities might be slightly deceiving, and that these values should mainly be used in comparative experiments, and not in absolute terms.

Figure 2: Actin mesh and normalized mesh densities measured in control cells and cells overexpressing SpireB from females fed with DMSO or Latrunculin A (LatA). The reduction in F-actin levels upon LatA feeding is similar in cells of either genotype, indicating that expression of SpireB does not alter the stability of actin filaments, but rather increases their abundance.

In addition, we have performed various experiments that further clarify the function of Capu and Spire:

a) Previously we used UTRN.GFP-expressing cells as controls for the *nos-Gal4,sqh-UTRN.GFP>UAS-Spire.B* cells (the SpireB overexpressing cells). We have now analyzed a better control, which are *nos-Gal4,sqh-UTRN.GFP>UAS-RFP* cells. We found that this genetic background already leads to a reduction in both persistent and diffusive motion (Table 1), but did not induce any detectable phenotypes. Therefore, we compared our SpireB and Capu overexpression experiments to this new control, and found that, although our initial observation holds still true, the reduction in flows speeds and active diffusion are less drastic (Table 1).

b) Instead of comparing our results to already published work on Capu overexpression, we repeated the overexpression of Capu and measured motion with our techniques. We found that Capu overexpression affects the oocyte in a similar way to SpireB overexpression. In both, cells over-expressing SpireB or Capu, cytoplasmic F-actin levels increase by ~1.5 fold, while active diffusion drops, and flows are not detectable. Table 1 and the appropriate parts of the manuscript (page 12 and 13) have been modified accordingly to our new findings. From all these experiments together, we conclude that the increase of F-actin in the cytoplasm negatively affects motion. We have also added a possible explanation to why our findings when Capu is overexpressed seem to differ from previous findings:

“Although this finding seems in contradiction to recent results³, the temporal resolution, the method for analyzing the motion and the level of detail in the analysis are different between the two studies”.

For the Capu over-expression we found:

$$D_{\text{ves,CapuOE}} = (1.2 \pm 0.7) \times 10^{-3} \mu\text{m}^2/\text{s} \quad D_{\text{act,CapuOE}} = (2 \pm 1) \times 10^{-3} \mu\text{m}^2/\text{s}$$
$$V_{\text{ves,CapuOE}} = \text{non detectable} \quad V_{\text{act,CapuOE}} = \text{non detectable}$$

c) Since the over-expression of Capu resulted in the same drop in motion as observed when SpireB is over-expressed, we concluded that variations in filament stability are probably not involved in the observed phenotype, and removed this statement from the manuscript. In fact, feeding LatA destroys the mesh in control cells and reduces the measured fluorescence intensities by around a factor of 1.5, both in control and in SpireB-overexpressing cells (Figure 2 above). This indicates that the stability of the actin filaments in SpireB-overexpressing cells is actually similar to that in control cells, while the higher fluorescence intensities measured are due to the fact that the cells contain more cytoplasmic F-actin in general.

(3)... I think the authors could test it, and furthermore, simply increase F-actin concentration by Jasplakinolide treatment to see whether the number of actin filaments affects cytoplasmic flow and active diffusion.

Since we now found that increasing both Capu and Spire levels in the oocyte results in an increase in the number of actin filaments, as well as a decrease in flows and diffusion, we did not feel the need for testing further how higher actin stability (by Jasplakinolide treatment in this case) affects motion.

(3)... In addition, I wonder whether Arp2/3 is involved in the actin mesh formation/dynamics, and thus active diffusion. This can be easily tested in the authors' setting with DDM when Arp2/3 is inhibited by CK-666.

We agree that the role of the Arp2/3 complex in cytoplasmic actin metabolism in the oocyte is rather elusive and in general constitutes an important question to be investigated.

We have performed various experiments trying to address this point:

a) We analyzed oocytes mutant for Arpc1 (a subunit of the Arp2/3 complex, *Arpc1*^{Q25sd} – null allele). We found that the actin mesh is formed in *Arpc1* mutant cells, but shows an aberrant morphology. DIC-DDM of vesicle movement showed a drop both in active diffusion and in flows speeds, indicating that Arp2/3 is involved in intracellular motion in the oocyte (page 13, line 438, Table 1). However, since we have no data regarding the mechanism of how Arp2/3 is involved in actin nucleation in the oocyte, we cannot at the moment explain the role of the Arp2/3 complex beyond our findings. We think that the concrete mechanism behind the involvement of the Arp2/3 complex in mesh dynamics and motion needs to be further investigated, and it is beyond the scope of this manuscript.

For Arpc1 mutant cells we found:

$$D_{\text{ves,Arpc1}} = (1.5 \pm 0.5) \times 10^{-3} \mu\text{m}^2/\text{s} \quad (\text{compared to controls } 3 \pm 1 \times 10^{-3} \mu\text{m}^2/\text{s})$$
$$V_{\text{ves,Arpc1}} = 12 \pm 8 \text{ nm/s} \quad (\text{compared to controls } 36 \pm 15 \text{ nm/s})$$

b) We treated cells *ex vivo* with two different concentrations of CK-666 (following published values), and measured intracellular motion of vesicles and F-actin. In our hands, this treatment did not cause any significant defects in motion, and could indicate that there is no acute function of Arp2/3 in regulating cytoplasmic motion.

c) We also tested an Arp3 RNAi, that resulted in egg chambers with cytokinesis defects in the nurse cells, and more than one oocyte per egg chamber. This strong phenotype in oogenesis prevented us from analyzing further these oocytes. We were unsuccessful in obtaining germline clones of a mutant allele of SCAR.

4. The manuscript presented an almost perfect correlation of vesicle diffusion and actin diffusion: $D_{act} \approx 2D_{ves}$. And the authors proposed that this is due to the vesicle size is around 2 times of the actin mesh size. It is somehow surprising that the ooplasmic vesicles come in such a uniform size. The oocyte contains lipid droplets, organelles, mRNA, and proteins that, to my understanding, may be of various diameters. Here I am curious whether the authors could test this hypothesis by examining vesicles of different sizes. For example, mitochondria undergo active fusion and fission, and display heterogeneity in size. It would be nice if the authors could perform the DDM analysis of mitochondria (by Mitotracker dye labeling, etc.) and actin mesh (by UTRN-GFP), to see whether the size of mitochondria does affect its diffusion rate.

Clearly, as pointed out by the reviewer, vesicles and other cytoplasmic components are far from being uniform in size. It is true that, in our DIC images, most of the optical signal is originated from approximately spherical particles with a size distribution peaked around 1 μm , and for this subfamily of vesicles we were able to provide a precise characterization of the motion.

We have now made clearer that the vesicles we describe along the paper are DIC-imaged vesicles (or DIC vesicles). We have also stated that "...Of note, our DIC imaging does not reveal all vesicles of the cell, as it is known that vesicles in the oocyte usually exhibit a wider size variation⁴ (page 10, line 348).

Following the Reviewer's advice, we have also generated movies of cells expressing a yellow fluorescent protein (YFP) targeted to the outer mitochondrial membrane (page 10, line 349). DDM analysis provided the following results for the effective diffusivity and streaming velocity of the mitochondria, respectively:

$$D_{\text{mito}} = (1.8 \pm 0.4) \times 10^{-3} \mu\text{m}^2/\text{s}$$

$$V_{\text{mito}} = (15 \pm 7) \text{ nm/s}$$

Unfortunately, simultaneous imaging of the actin mesh was not possible, because no "red" (far enough from YFP) F-actin probe that labels the actin mesh is available. We tested the commonly used LifeAct.mCherry and F-Tractin.tdTomato, and found that although both can be expressed in the germline, they fail to label the actin mesh. Therefore, we concentrated on analyzing mitochondrial motion only.

As anticipated by the Reviewer, the mitochondria appear to be highly heterogeneous in size and shape (Movie S3). This made it difficult to assess a characteristic size of the structure. The average size of a mitochondrion at mid-oogenesis is $\sim 0.8 \times 0.2 \mu\text{m}$, with a maximum length of 1.4 μm ⁵. This value is significantly lower than the characteristic size of the vesicles

imaged by DIC. However, the observed diffusivity D_{mito} is lower than the diffusivity $D_{\text{ves}} = (3 \pm 1) \times 10^{-3} \mu\text{m}^2/\text{s}$ of the DIC vesicles. This finding contradicts the tentative argument provided in the first version of the manuscript to explain the experimental finding $D_{\text{act}} \approx 2D_{\text{ves}}$. Nevertheless, mitochondria are known to form highly organized networks and are subject to fusion and fission events, clearly arguing for a strong interaction between them. So far it is unknown whether - and how - organelle-organelle interactions affect the motion of organelles, while it is reasonable to assume that such processes could slow down intracellular movement.

Retrospectively, we acknowledge that our argument was too naïve. In fact, while the assumption that the effective diffusion coefficient of a *passive* probe particle is inversely proportional to its size is reasonable - and has been experimentally confirmed (see for example⁶ - this is far from being obvious for cellular components. These components, like vesicles and mitochondria, are known to have *specific* interactions with the cellular environment and with the F-actin network in particular.

As a consequence of these new findings and arguments, we decided to remove the tentative argument to explain the experimental finding $D_{\text{act}} \approx 2D_{\text{ves}}$. We are very grateful to the Reviewer for her/his suggestion.

Reviewer 2:

The major claim of the paper is that differential dynamic microscopy (DDM) is a potent tool to understand passive and active transport in complex environments, specifically in drosophila oocyte cells. This reviewer's experience with DDM suggests that the claim is reasonable: the method lets its user obtain dynamic information even in environments that would be difficult to explore by traditional methods (e.g., dynamic light scattering, DLS) and with better precision than most particle tracking/velocimetry algorithms (which additionally require the particles to be resolved or at least distinguishable against the background, unlike DDM).

We thank the reviewer for the positive feedback about our work, especially given her/his experience with DDM, a technique that is undoubtedly becoming more popular, but it is still unknown to a large public.

I wish figure 1D (Γ vs q in a log-log representation) would be given one or two linear-linear insets; for the velocity motion, plot Γ vs q ; for the diffusion Γ vs q^2 . Even better might be Γ/q and Γ/q^2 vs q and q^2 respectively...should be flat lines, showing off how well the method really works.

Following the reviewer's suggestion, we added a panel in figure 1D reporting the ratios $\Gamma_1(q)/(Dq^2)$ and $\Gamma_2(q)/(v_0q)$, where D and v_0 are constants obtained from the fit of Γ_1 and Γ_2 , respectively. As pointed out also by the reviewer, this representation of the data provides a quite stringent test of the accuracy of the method and of the adopted model. The fact that the obtained curves do not show any systematic deviation and an overall moderate fluctuation (<10%) with respect to the mean value 1, provides a convincing evidence of the reliability of our approach.

Not being an expert in microbiology, I am unable to comment on another claim of the paper, which is that the particular cells chosen for investigation need to be studied to strengthen claims made for

mammalian oocytes, but this hardly matters because this paper will advance an important tool for a wide variety of studies.

We thank the reviewer for acknowledging the methodological relevance and the generality of our work.

The presentation is logical, and I found only minor grammatical and mechanical flaws with the writing. I imagine the copy editors will repair these and others:

Line 39: progresses  progress

Line 76: require  requires

Line 121: contributes  contributions

Line 143: Under  From

All the typing errors reported by the reviewer have been corrected.

- 1 Röper, K. & Brown, N. H. Maintaining epithelial integrity: a function for gigantic spectraplakins isoforms in adherens junctions. *The Journal of cell biology* **162**, 1305-1315, doi:10.1083/jcb.200307089 (2003).
- 2 Nashchekin, D., Fernandes, A. R. & St Johnston, D. Patronin/Shot Cortical Foci Assemble the Noncentrosomal Microtubule Array that Specifies the Drosophila Anterior-Posterior Axis. *Dev Cell* **38**, 61-72, doi:10.1016/j.devcel.2016.06.010 (2016).
- 3 Bor, B., Bois, J. S. & Quinlan, M. E. Regulation of the formin Cappuccino is critical for polarity of Drosophila oocytes. *Cytoskeleton (Hoboken)* **72**, 1-15, doi:10.1002/cm.21205 (2015).
- 4 Compagnon, J., Gervais, L., Roman, M. S., Chamot-Boeuf, S. & Guichet, A. Interplay between Rab5 and PtdIns(4,5)P2 controls early endocytosis in the Drosophila germline. *J Cell Sci* **122**, 25-35 (2009).
- 5 Tourmente, S., Lecher, P., Degroote, F. & Renaud, M. Mitochondrial development during Drosophila oogenesis: distribution, density and in situ RNA hybridizations. *Biol Cell* **68**, 119-127 (1990).
- 6 Fodor, E. *et al.* Active-driven fluctuations in living cells. *EPL (Europhysics Letters)* **110**, 48005 (2015).

REVIEWERS' COMMENTS:

Reviewer #1 (Remarks to the Author):

I appreciate that the authors made the great efforts to address all my questions/concerns. The changes in responding to the questions/concerns have been incorporated nicely in the revised manuscript. I would recommend it be accepted for publication in Nature communications.